# Frequent *FOXA1*-Activating Mutations in Extramammary Paget’s Disease

**DOI:** 10.3390/cancers12040820

**Published:** 2020-03-29

**Authors:** Takuya Takeichi, Yusuke Okuno, Takaaki Matsumoto, Nobuyuki Tsunoda, Kyogo Suzuki, Kana Tanahashi, Michihiro Kono, Toyone Kikumori, Yoshinao Muro, Masashi Akiyama

**Affiliations:** 1Department of Dermatology, Nagoya University Graduate School of Medicine, Nagoya 466-8550, Japan; midnightexpress0531@hotmail.com (T.M.); tanahashi@med.nagoya-u.ac.jp (K.T.); miro@med.nagoya-u.ac.jp (M.K.); ymuro@med.nagoya-u.ac.jp (Y.M.); makiyama@med.nagoya-u.ac.jp (M.A.); 2Medical Genomics Center, Nagoya University Hospital, Nagoya 466-8550, Japan; yusukeokuno@gmail.com; 3Department of Breast and Endocrine Surgery, Nagoya University Graduate School of Medicine, Nagoya 466-8550, Japan; nobtsun@med.nagoya-u.ac.jp (N.T.); kikumori@med.nagoya-u.ac.jp (T.K.); 4Department of Pediatrics, Nagoya University Graduate School of Medicine, Nagoya 466-8550, Japan; kyogo-s@med.nagoya-u.ac.jp

**Keywords:** extramammary Paget’s disease, forkhead box A1, fusion gene, mammary Paget’s disease, somatic mutations, whole-genome sequencing

## Abstract

Extramammary Paget’s disease (EMPD) is a neoplastic skin disease of indeterminate origin with an unknown genetic cause. We performed a comprehensive genetic analysis or targeted gene sequencing in 48 patients with EMPD. We identified *FOXA1* mutations, a *GAS6–FOXA1* fusion gene, and somatic hotspot mutations in the *FOXA1* promoter region in 11 of the 48 EMPD patients (11/48, 23%). Additional mutations were identified in *PIK3CA* (six patients) and in *HIST1H2BB*, *HIST1H2BC*, and *SMARCB1* (one patient each), but none were found in other frequently mutated genes in cancer. A global gene expression analysis using EMPD clinical samples found the upregulation of PI3 kinase–AKT–mTOR signaling. *ABCC11*, which is specifically expressed in the apocrine secretory cells and is necessary for their sweat secretion, was upregulated in the EMPD samples. This upregulation suggests that Paget cells originate from apocrine secretory cells. Immunohistochemical staining revealed that FOXA1 expression was prevalent in all of the EMPD samples analyzed and was associated with estrogen receptor expression. Our genetic analysis indicates that EMPD frequently involves *FOXA1* mutations. FOXA1 is a transcriptional pioneer factor for the estrogen receptor, and the present results suggest that certain treatments for hormone-dependent cancers could be effective for EMPD.

## 1. Introduction

Paget’s disease (PD) is a neoplasm seen on the nipple/areola (mammary PD; MPD) or in extramammary body zones, such as in anogenital and perineal skin and the axilla (extramammary PD; EMPD) [1]. MPD is relatively rare, observed in 0.7–4.3% of all breast cancers. It is much more frequent in women, possibly because of the clear predominance of breast cancer in females (sex ratio of 1:50–200). MPD develops more often in people in their fifties (mean age: 57 years), i.e., in 70% of MPD cases in postmenopausal women, but it has been observed in adolescents and in elderly patients [1]. EMPD was first described by Crocker in 1889 [2]. It shares several clinicopathological similarities with its mammary homologue but shows some differences, namely, pathogenesis [1]. The precise incidence of EMPD is unknown, although it is possibly rarer than MPD, accounting for 6.5% of all reported cases of PD. It predominantly affects patients aged 65 to 70 years, and 90% of EMPD patients are older than 50 years [1]. In Asian populations, a male predominance for EMPD is observed [3], although there is a clear female preponderance of EMPD in non-Asian populations.

Surgical resection is the first choice of treatment for resectable cases; once PD becomes unresectable, it is difficult to control by chemotherapy, and the prognosis is poor. For systemic metastases, including extensive lymph node metastases, no systemic chemotherapy improves the patient’s survival [4]. Moreover, as most patients with EMPD are elderly, the option of systemic chemotherapy is frequently limited [1,4]. Thus, invasion levels and the presence or absence of multiple lymph node metastases are important prognostic factors in EMPD [4].

Although cutaneous EMPD usually occurs in skin regions that are rich in apocrine glands, its exact cellular origin has yet to be determined. It has been proposed that EMPD arises from skin pluripotent stem cells [5]. Toker cells, which are a small population of benign pagetoid clear cells in normal nipples, have also been proposed as a candidate for the origin of Paget cells [6].

Several sequencing studies using targeted or whole-exome sequencing approaches have revealed that EMPD and MPD carry driver mutations in *PIK3CA*, *KRAS*, *BRAF*, *AKT1*, and other genes [7,8,9,10]. Although the aforementioned genes are frequently mutated in various cancers [11] and are not particularly characteristic of PD, it is important to gather information on these candidate driver mutations towards understanding the underlying molecular genetic basis of EMPD. In this study, we performed a comprehensive genetic analysis, including whole-genome sequencing, of our series of EMPD patients and a targeted sequencing analysis of MPD patients to elucidate the molecular pathogenesis of EMPD.

## 2. Results

### 2.1. Whole-Genome Sequencing of Two Patients with EMPD

We performed whole-genome sequencing using frozen specimens containing Paget cells and peripheral blood mononuclear cells derived from two patients with EMPD (Figure 1, Appendix A). In one patient (UPN1), we identified a novel in-frame fusion involving *GAS6* (encoding growth arrest-specific protein 6) and *FOXA1* (encoding forkhead box A1 or hepatocyte nuclear factor 3α), along with 43 nonsynonymous somatic point mutations (Figure 1A). The *GAS6–FOXA1* fusion gene identified in UPN1 was generated by a balanced translocation between chromosomes 13 and 14 (Figure 1C and Appendix A). This translocation connected the initial two exons and the second intron of *GAS6* to a point 10 kb upstream of *FOXA1*, which was validated by both Sanger sequencing and RNA sequencing (Figure 1D). The resulting transcript contained exons 1 and 2 of *GAS6* and exon 2 of *FOXA1*, whose combination resulted in an in-frame fusion of the two proteins. While all functional domains of GAS6, except a truncated Gla domain, were lost in the fusion, the FOXA1 forkhead domain, the essential domain for its transcriptional activity, was conserved (Figure 1E). Based on the protein structure of GAS6–FOXA1, we considered that the function of this gene fusion is to upregulate the transcriptional activity of *FOXA1* using *GAS6* promoter activity.

In another patient (UPN2), we identified several possible driver mutations, including a mutation affecting the promoter region of *FOXA1* (g.chr14:38064406G>A in the hg19 genome coordinate), a *PIK3CA* (encoding phosphatidylinositol-4,5-bisphosphate 3-kinase catalytic subunit alpha) p.E81K mutation, and a *HIST1H2BC* (encoding histone cluster 1 H2B family member c) p.K24N mutation (Figure 1B). The *FOXA1* promoter mutation is located 81 bp upstream of the gene’s transcription start site (Figure 1F and Appendix A) and has been reported to upregulate *FOXA1* expression [12]. The identical *FOXA1* promoter mutation is reported to be a hotspot mutation in breast cancer, albeit mutated in <1% of patients, and is known to upregulate the transcription of this gene and to give a growth advantage to breast cancer cells under anti-estrogen receptor therapy in vitro. The *PIK3CA* p.E81K mutation is also a known hotspot mutation in cancer and is recurrently identified in an inherited disease (*PIK3CA*-related overgrowth spectrum) [13]. The *HIST1H2BC* p.K24N mutation was not reported in the literature nor in the Catalogue Of Somatic Mutations In Cancer (COSMIC) database (https://cancer.sanger.ac.uk/cosmic/accessed at 04/10/2019), although the affected amino acid residue is a known target of histone acetylation [14].

Combined, the whole-genome analysis found *FOXA1* to be affected in both patients. Somatic copy number abnormalities were not detected in these patients.

### 2.2. Whole-Exome Sequencing of 21 Patients with EMPD

We performed exome sequencing in 21 additional patients with EMPD (UPN11–UPN21 and UPN39–UPN48 in Appendix A) (Figure 2, Appendix A). We identified a total of 428 somatic point mutations (0–77 mutations per patient). The other recurrently mutated gene was *PIK3CA* (four mutations in three patients, Figure 2A). Other possible driver mutations were identified in one patient each (*HIST1H2BB*, *HIST1H2BC*, and *SMARCB1*).

### 2.3. RNA Sequencing of Six Patients with EMPD

We performed RNA sequencing in six additional patients with EMPD (UPN3 and UPN5–9 in Appendix A) for whom RNA of sufficient quality was available. A patient (UPN9) carried a fusion gene involving *PDIA5* (encoding the protein disulfide isomerase family A member 5) and *TMEM45A* (encoding the transmembrane protein 45A) (Figure 2B). The predicted protein structure included the signal peptide and thioredoxin domains 1–2 (and part of domain 3) of PDIA5, an inserted isoleucine residue, and all of TMEM45. However, the driver role of this fusion gene is unclear, to the best of our knowledge.

### 2.4. Targeted Sequencing in 48 Patients with EMPD or MPD

Finally, we performed a targeted sequencing study that included mutated genes found in whole-genome/exome sequencing studies, genes frequently mutated in all cancers, and *FOXA1* and the 0–200 kb upstream region in all 48 patients with EMPD and 14 patients with MPD (Figure 2, Appendix A).

We identified the recurrent *FOXA1* promoter mutation (nine g.chr14:38064406G>A and one g. chr14:38064406G>T mutation) in a total of 10 patients with EMPD. The other recurrent mutations were seven *PIK3CA* mutations found in six patients. Four of the seven *PIK3CA* mutations affected glutamic acid (E) residues, and there was a mutational hotspot at the beginning of the helical domain (Figure 2C). This distribution of mutations is similar to that observed in several cancers [15]. Other mutations were identified in one patient each, although *HIST1H2BB* and *HIST1H2BC* were at the identical amino acid residue (Lys24), which is a known target of acetylation, suggesting a redundant role of these two mutations (Figure 2D,E). A frameshift mutation in *SMARCB1* is commonly observed in pan-cancer analysis (Figure 2F) [11]. *TP53*, *ARID1A*, *PTEN*, and *KRAS*, which are frequently mutated in skin or breast cancers, were not mutated in our EMPD cohort. We did not identify fusion genes involving *FOXA1* by this targeted sequencing in addition to *GAS6–FOXA1*. 

We identified a *PIK3CA* p.H1047R mutation in a patient with MPD. The difference between EMPD and MPD in terms of mutations is inconclusive: the frequencies of the *FOXA1* promoter mutations showed no statistically significant differences (10/48 vs. 0/14, *p* = 0.0987).

### 2.5. Global Gene Expression Profiling of EMPD

We compared global gene expression profiles of EMPD (*n* = 8, UPN1–3 and UPN5–9) to those of normal skin tissue samples obtained from healthy volunteers (*n* = 5) (Figure 3, Appendix A). Unsupervised hierarchical clustering clearly distinguished EMPD samples from normal skin samples, with a bootstrap probability of 100% (Figure 3A). In the patients with EMPD, sub-clusters showed no correlation with disease severity nor with the mutational status of *FOXA1* or *PIK3CA*. We performed a gene set enrichment analysis using the hallmark gene set database [16]. In the EMPD samples, genes associated with PI3 kinase–AKT–mTOR signaling (HALLMARK_MTORC1_SIGNALING) and estrogen response (HALLMARK_ESTROGEN_RESPONSE_EARLY) were upregulated (Figure 3B). EMPD samples also showed the upregulation of genes associated with the malignant transformation of several cancers, including luminal breast cancer (Appendix A). 

When several individual genes were focused on, *FOXA1* expression showed statistically significant upregulation in EMPD samples (*p* = 9.6 × 10^−27^, the 51st highest upregulation in global expression profiling) (Figure 3C). We also found that *ABCC11*, which encodes the ATP-binding cassette sub-family C member 11 and is required for apocrine sweat secretion, was the seventh statistically significant upregulated gene (*p* = 1.3 × 10^−51^, the seventh highest upregulation in global expression profiling) (Figure 3D). Among various cell types in skin tissue, *ABCC11* is specifically expressed in apocrine secretory cells [17]. In addition to *FOXA1* and *ABCC11*, genes encoding keratin family proteins were differentially expressed in the EMPD samples (Figure 3E). *KRT8*, *KRT18*, and *KRT19*, which are expressed in human eccrine sweat cells [18], were upregulated in the EMPD samples. *KRT2* and *KRT15*, which are associated with suprabasal cells and stratified squamous epithelia, respectively [19], were downregulated in the EMPD samples.

### 2.6. Immunostaining in EMPD/MPD Samples

We performed immunohistochemical staining for FOXA1. As FOXA1 is known to assist the transcriptional activity of the estrogen receptor (ER) and may work cooperatively in the tumorigenesis of breast cancer [20], we also performed immunohistochemical staining for ER. FOXA1 was found to be strongly expressed in the nuclei of Paget cells in all the examined cases of EMPD (UPN1–UPN48 except for UPN39), regardless of the mutational status of the *FOXA1* promoter or the presence or absence of the *GAS6–FOXA1* fusion gene (Figure 4E,H,K). The staining intensity was similar in all samples with or without the *FOXA1* promoter mutations. In contrast, FOXA1-positive cells were almost completely absent in 10 age-matched normal control skin samples (Figure 4B).

ER staining was positive in 30/46 (65%) of the EMPD cases and in 2/12 (17%) of the MPD cases, with varied staining intensities, according to the Allred score (Appendix A).

## 3. Discussion

In this study, we identified the *GAS6–FOXA1* fusion in an EMPD case and recurrent hotspot mutations in the *FOXA1* promoter in 10 EMPD patients. This is the first report of a fusion gene in EMPD, and to the best of our knowledge, the combination of *GAS6* and *FOXA1* is a novel fusion pair. Based on the structure of the fusion gene, we speculate that the *GAS6*–*FOXA1* fusion might lead to FOXA1 protein expression resulting from the use of the promoter region of *GAS6*. This hypothetical mechanism is not exactly a “promoter hijack”, but it may hijack the promoter and the signal peptide domain en bloc. Indeed, the tumor tissue of UPN1 showed higher FOXA1 expression at both the mRNA level and the protein level (Figure 3 and Appendix A).

The hotspot mutation in the *FOXA1* promoter region, in addition to mutations within the coding sequence, has been reported in breast cancer and leads to the overexpression of *FOXA1* through increased E2F binding [12]. The frequency of the promoter mutation is apparently higher in our EMPD cohort (21%) than in breast cancer (<1%), suggesting the importance of FOXA1 upregulation in EMPD. In addition, all of the analyzed cases exhibited strong expression of this gene at both the transcript level and the protein level. The strong immunostaining of the FOXA1 protein in EMPD has also been reported by another study group [21]. The prevalence of *FOXA1* upregulation in EMPD suggests the gene’s non-redundant role in the molecular pathogenesis of this disease and indicates the possibility that undiscovered molecular mechanisms which also upregulate *FOXA1* expression exist, other than genetic defects in *FOXA1* itself. Such mechanisms may include genomic alterations that affect other regulatory elements of *FOXA1* and/or epigenetic abnormalities that affect *FOXA1* expression [22].

Global expression profiling identified the strong differential expression of *ABCC11* in the EMPD samples. The gene is reported to be specifically expressed in apocrine secretory cells among cells that organize the skin tissue [17]. Combined with several indirect items of evidence, such as the preferential occurrence of PD in skin regions with abundant apocrine glands and the histopathological appearance of PD as an adenocarcinoma, our results strongly indicate that Paget cells originate from apocrine secretory cells.

Mai et al. reported the co-expression of FOXA1 and ER in 95% (18/19) of ER-positive PD cases [21]. Our ER staining results are similar to those of immunohistochemical analyses in that previous report. FOXA1 is a transcriptional pioneer factor that opens chromatin and allows the ER to access its genomic targets [23]. High FOXA1 levels have been observed in tumors with poor prognoses and in breast cancer metastases, where FOXA1 overexpression reprograms the ER binding landscape [23]. Rheinbay et al. generated MCF-7 cells that stably overexpress FOXA1 and treated them with the ER antagonist fulvestrant, a compound used to treat patients with hormone-receptor-positive breast cancer [12]. FOXA1-overexpressing cells grow significantly faster than control cells even under fulvestrant treatment, suggesting that increased FOXA1 expression induces cellular tolerance to anti-ER treatment in breast cancer [12]. Furthermore, *FOXA1* is mutated in 3–5% of prostate cancers and in subsets of cancers that exhibit the amplification of the genomic region encompassing the *FOXA1* gene [24]. The FOXA1 protein is required for epithelial cell differentiation in the mouse prostate [25] and promotes cell cycle progression in castration-resistant prostate cancer [26]. Thus, *FOXA1* mutations could be associated with the pathogenesis of hormone-dependent cancers. 

## 4. Materials and Methods

### 4.1. Patients

We studied frozen tumor specimens from 20 EMPD patients (8 men and 12 women, UPN1–UPN10, UPN39–UPN48 in Appendix A) and formalin-fixed, paraffin-embedded (FFPE) tissue from 28 EMPD patients (18 men and 10 women, UPN11–UPN38 in Appendix A, mean age: 74.2 years; range: 55–92 years; primary location: 2 axillary, 1 inguinal, 20 penoscrotal, 15 perineal). Detailed clinical information of the EMPD patients is summarized in Appendix A. In addition, we analyzed 14 FFPE tissue samples from 14 female patients with MPD. We diagnosed EMPD and MPD from characteristic histopathological findings of the affected tissues, excluding Paget’s phenomena of adenocarcinomas derived from other organs. Differential diagnoses were ruled out by Alcian blue and periodic acid–Schiff (PAS) staining and immunohistochemical analyses (CK7, CAM5.2, and GCDFP15). We obtained written informed consent from the patients. The Ethics Committee of the Nagoya University Graduate School of Medicine approved this study (the ethical code: 2018-0159), which was conducted in accordance with the principles of the Declaration of Helsinki. Skin samples from five healthy individuals were used as controls (mean age: 67.0 years; range: 53–87 years).

### 4.2. Nucleic Acid Preparation

We extracted genomic DNA from frozen specimens and peripheral blood mononuclear cells using the QIAamp DNA Blood Mini Kit (Qiagen, Hilden, Germany), according to the manufacturer’s instructions. We also extracted genomic DNA from FFPE tissues by using the GeneRead DNA FFPE Kit (Qiagen). We obtained total RNA from frozen specimens by the RNeasy Mini Kit (Qiagen). The quality of the extracted RNA was assessed using RNA ScreenTape and the Agilent 2200 TapeStation system (Agilent, Santa Clara, CA, USA).

### 4.3. Whole-Genome Sequencing

Whole-genome sequencing (WGS) was performed using the HiSeq X platform (Illumina, San Diego, CA, USA) to obtain 2 × 150-bp paired-end reads to cover the human genome with a mean coverage of 40×. The reads were aligned to the hg19 reference genome using the Burrows–Wheeler Aligner with default parameters and a –mem option (http://bio-bwa.sourceforge.net/). PCR duplicates were removed using Picard tools (http://broadinstitute.github.io/picard/). We used VarScan2 (http://varscan.sourceforge.net/) to detect somatic nucleotide alterations with <0.0001 *p*-value (provided by VarScan2), >0.1 variant allele frequency (VAF) in the patient’s tumor sample, <0.05 VAF in the patient’s germline sample, and an average of <0.01 VAF in germline samples from 12 unrelated healthy donors. Chromosomal copy numbers were simply estimated from the number of reads in a 10 kb bin of each chromosomal location. Candidates of chromosomal structural variations were identified by re-aligning soft-clipped nucleotide sequences using BLAT (https://hgwdev.gi.ucsc.edu/~kent/src/) with default parameters and a –stepSize = 5 option. The called candidates supported by three or more reads were inspected using integrative genomics viewer (https://software.broadinstitute.org/software/igv/) and were validated by Sanger sequencing. The results were visualized using Circos [27].

### 4.4. Whole-Exome Sequencing 

We performed whole-exome sequencing using the SureSelect XT Target enrichment system and the SureSelect Human All Exon V6 bait (Agilent). We obtained 2 × 150-bp paired-end sequencing data using a HiSeq2500 next-generation sequencing platform to cover the exome with 100× coverage, and variant calling was performed essentially in the same way as whole-genome sequencing. 

### 4.5. Targeted Gene Sequencing

We designed a custom SureSelect target enrichment bait that covered all of *FOXA1* and its 0–200 kb upstream region, all of *GAS6*, the mutant genes in our whole-genome sequencing study, and common mutant genes in cancers (Appendix A). Variant calling and structural variation detection were performed essentially in the same way as whole-genome sequencing. We also performed PCR-based targeted deep sequencing covering the mutational hotspot in *FOXA1* promoter. The primer sequences are available upon request.

### 4.6. RNA Sequencing

We prepared sequencing libraries using the NEBNext Ultra II RNA Prep Kit with the NEBNext Poly(A) mRNA Magnetic Isolation Module (New England Biolabs, Ipswich, MA, England). A 2 × 100-bp paired-end sequencing was performed to obtain 60 million reads per sample. The data were aligned using Tophat2 (https://ccb.jhu.edu/software/tophat/index.shtml) with default parameters, and putative fusion genes were called using TopHat-fusion (https://ccb.jhu.edu/software/tophat/fusion_index.shtml) with default parameters. The expression level of each gene was calculated from the number of reads on exons using HTSeq (https://htseq.readthedocs.io/). Differential expression was calculated using DESeq2 (https://bioconductor.org/packages/release/bioc/html/DESeq2.html). Hierarchical clustering was performed using pvclust (http://stat.sys.i.kyoto-u.ac.jp/prog/pvclust/), which is a package of R (https://www.r-project.org/) that performs various clustering analyses with bootstrapping. Gene set enrichment analysis was performed using the Molecular Signature Database (v6.1, http://www.broad.mit.edu/gsea/; accessed at 18/04/2019) based on the stat values obtained by DESeq2. We used the hallmark gene set database comprising 50 gene sets and considered a false discovery rate (FDR) <0.1 to be statistically significant. FOXA1 target genes and FOXA1 pathway genes were defined based on the TRANSFAC database (http://genexplain.com/transfac/) and the Pathway Interaction Database [28], respectively.

### 4.7. Immunohistochemistry

Immunohistochemical analysis of skin samples from the participants was performed as described previously, with slight modifications [29]. FOXA1 immunostaining was performed using anti-FOXA1 antibodies (clone [EPR10881], ab170933, Abcam, Cambridge, UK), which is a rabbit monoclonal antibody that recognizes the C-terminal amino acids of the protein. We also used anti-estrogen receptor (ER) alpha (clone [SP1], ab16660, Abcam) antibodies as the primary antibody, according to the manufacturer’s instructions. We assessed the staining patterns of ER according to the Allred score [30] and categorized the samples in three groups: ++ (Allred score >3), + (Allred score = 3), and – (Allred score <3).

### 4.8. Data Availability

The NGS data can be accessed at the DDBJ Sequence Read Archive (DRA) (https://www.ddbj.nig.ac.jp/dra/index-e.html).

## 5. Conclusions

Increased FOXA1 expression in EMPD cells might also cause cellular proliferation or survival similar to those seen in hormone-dependent cancers. FOXA1 is a transcriptional pioneer factor for the estrogen receptor, and our results suggest that certain treatments for hormone-dependent cancers might be effective against EMPD. Furthermore, developing molecular targeted therapies against dysregulated FOXA1 expression in EMPD is tempting, as the mutations are much more frequent than in breast cancers and the therapy may translate to a breast cancer therapy. Conversely, if a target therapy against FOXA1 upregulation in breast cancer is developed, it may be incorporated into PD chemotherapies, which are currently sub-optimal.

In conclusion, we found frequent *FOXA1* mutations in EMPD cases and showed that FOXA1 is upregulated in almost all EMPD lesions. Further genomic/epigenomic analyses focusing on *FOXA1* and the development of FOXA1-targeted therapies are warranted.

## Figures and Tables

**Figure 1 cancers-12-00820-f001:**
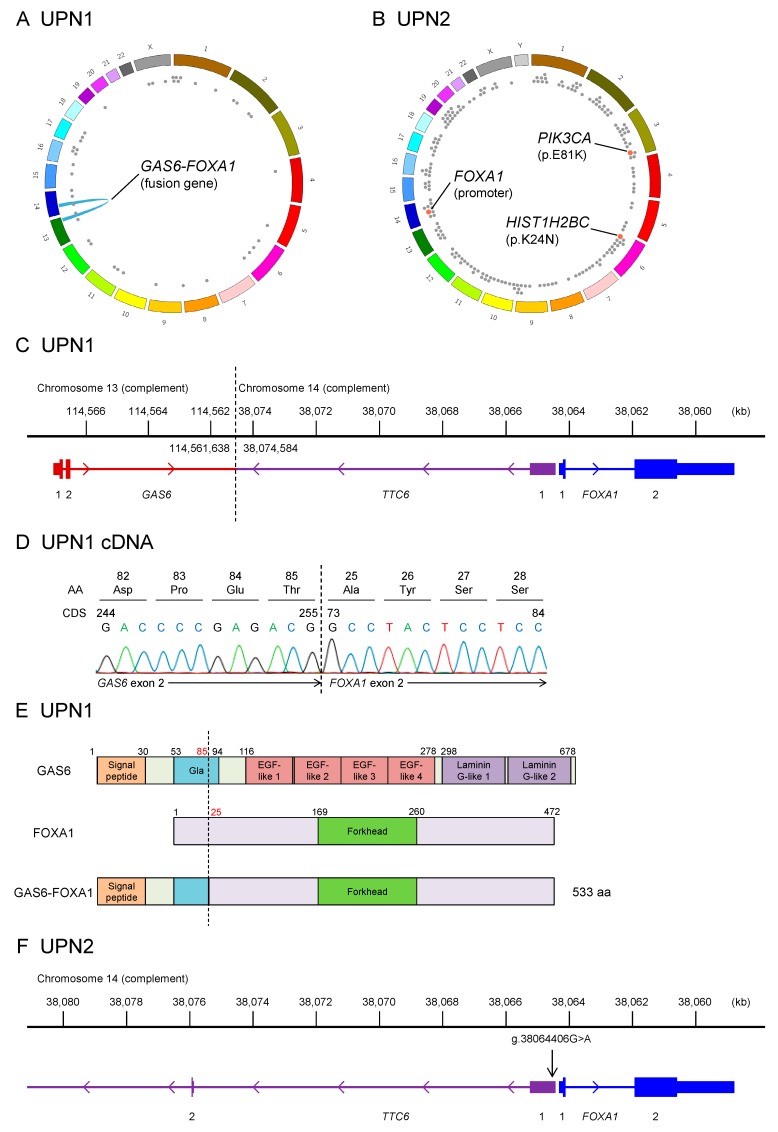
FOXA1-activating mutations in extramammary Paget’s disease (EMPD) identified by whole-genome sequencing. (**A**,**B**) Summary of somatic mutations identified in patients UPN1 (**A**) and UPN2 (**B**). Dots indicate nonsynonymous mutations, and the blue arch indicates gene fusion. We identified 43 somatic point mutations and a gene fusion of *GAS6* and *FOXA1* in UPN1. A total of 190 somatic point mutations were identified in UPN2, 3 of which were possible driver mutations. (**C**) Chromosomal structure of the *GAS6–FOXA1* fusion gene. Genome coordinates, transcripts, and the breakpoint (dashed line) are indicated. (**D**) Complementary DNA sequence of the *GAS6–FOXA1* fusion gene. Exon 2 of *GAS6* is joined to exon 2 of *FOXA1*, resulting in an in-frame fusion. (**E**) Predicted protein structure of GAS6–FOXA1. (**F**) Position of the *FOXA1* promoter mutation (g.38064406G>A), which is 81 bp upstream of the transcription start site of *FOXA1*.

**Figure 2 cancers-12-00820-f002:**
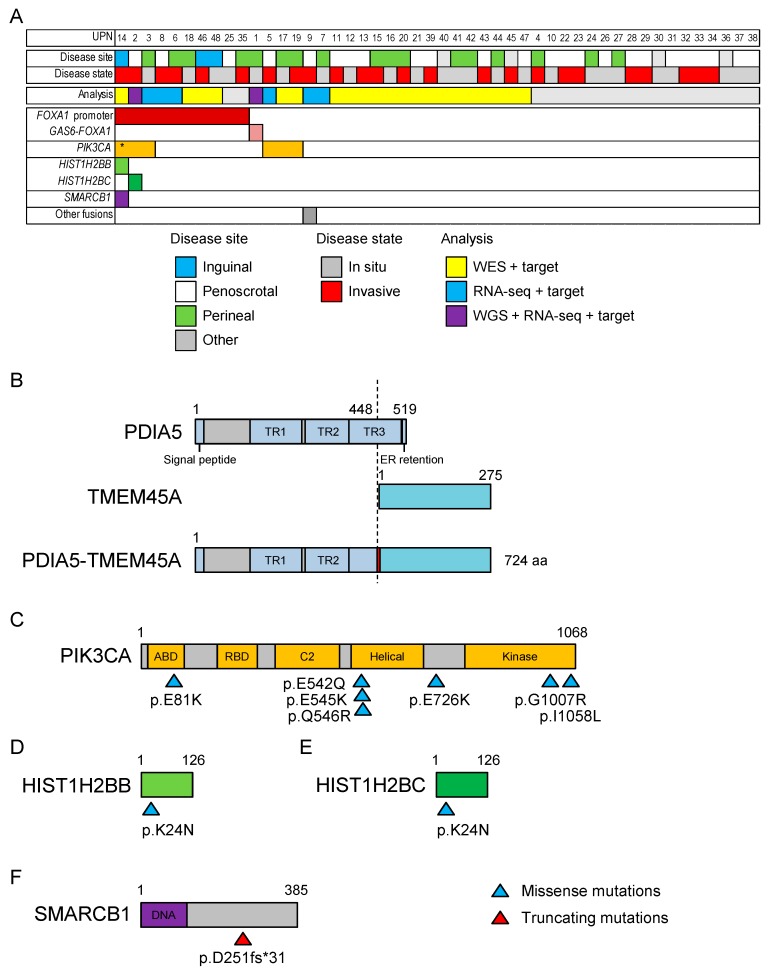
Summary of somatic mutations in EMPD. (**A**) The mutational landscape of EMPD. Yellow, blue, purple, and gray indicate patients analyzed by whole-exome sequencing (WES) plus targeted sequencing, RNA sequencing (RNA-seq) plus targeted sequencing, whole-genome sequencing (WGS) plus RNA-seq plus targeted sequencing, and targeted sequencing, respectively. Boxes with colors indicate the presence of mutations. An asterisk indicates two coexistent *PIK3CA* mutations within a patient. (**B**) Predicted protein structure of the *PDIA5–TMEM45A* fusion gene identified in UPN9. Amino acid residues 1–448 of PDIA5 and 1–275 of TMEM45A are connected by an additional isoleucine residue between them (indicated by red). TR, thioredoxin domain; ER, endoplasmic reticulum. (**C**–**F**) The distribution of somatic mutations in affected genes. Blue and red triangles indicate missense and truncating mutations, respectively. Numbers indicate amino acid numbers. ABD, p85-binding domain; RBD, Ras binding domain; C2, C2 PI3K-type domain; DNA, DNA binding domain.

**Figure 3 cancers-12-00820-f003:**
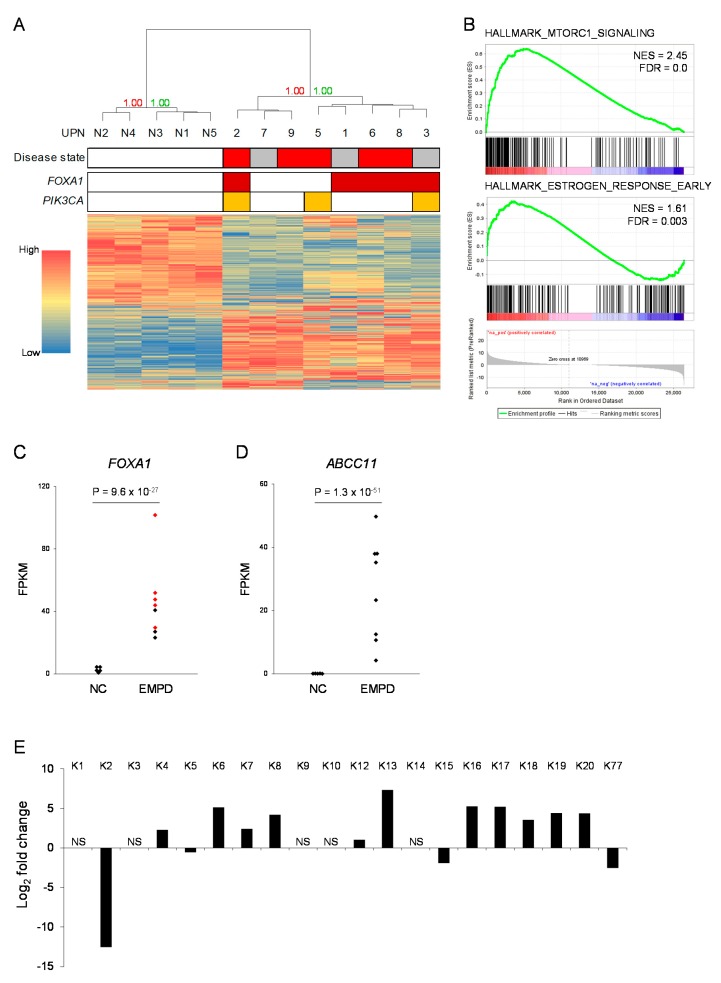
Global gene expression profiling of EMPD. (**A**) Hierarchical clustering using the global expression profiles obtained by RNA-seq. N1–N5 indicate normal skin samples obtained from healthy volunteers. On the disease state line, red and gray boxes indicate invasive disease state and in situ disease state, respectively. Red and green numbers indicate the bootstrap probability and the approximately unbiased *p*-value, respectively. (**B**) Gene set enrichment analysis using the hallmark gene set database. Genes associated with mTOR signaling (HALLMARK_MTORC1_SIGNALING) and estrogen response (HALLMARK_ESTROGEN_RESPONS_EARLY) are relatively upregulated in EMPD samples. NES, normalized enrichment score; FDR, false discovery rate. (**C**,**D**) Expression levels (fragments per kilobase per million; FPKM) of *FOXA1* (**C**) and *ABCC11* (**D**). Red dots in (**C**) indicate samples associated with *FOXA1* aberrations, NC, normal controls. (**E**) Differential expression of keratin family genes between EMPD and normal skin samples. Positive values indicate genes upregulated in EMPD samples. NS, not significantly different (adjusted *p*-value >0.1).

**Figure 4 cancers-12-00820-f004:**
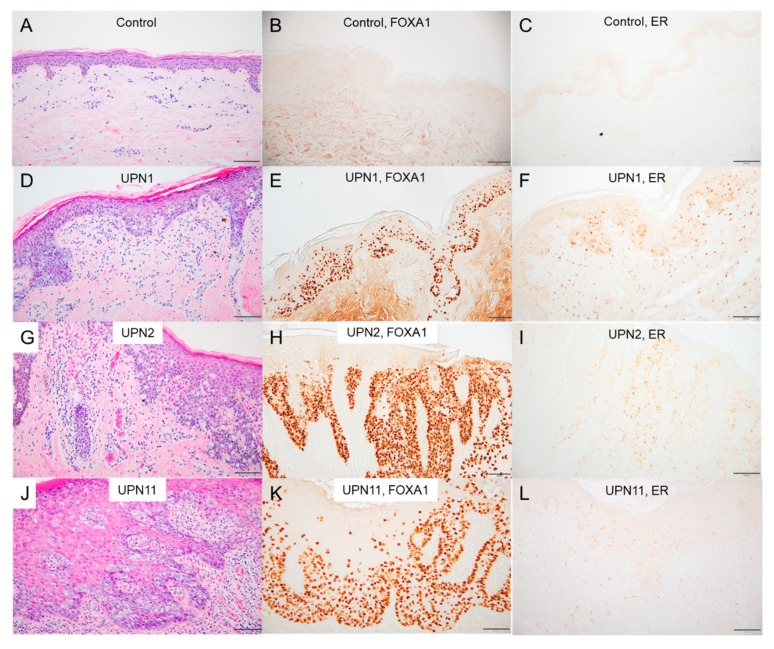
Immunohistochemical analyses of EMPD lesions and normal epidermis. Skin specimens from a healthy donor (**A**–**C**); UPN1, the case with the *GAS6-FOXA1* fusion (**D**–**F**); UPN2, the case with the *FOXA1* promotor mutation (**G**–**I**); and UPN11, a case with no detectable mutation (**J**–**L**) stained with anti-FOXA1 (**B**,**E**,**H**,**K**) and anti-estrogen receptor alpha (**C**,**F**,**I**,**L**) antibodies (scale bars, 100 μm).

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
