# Peer review of "Frequent FOXA1-Activating Mutations in Extramammary Paget’s Disease"

_cancers, 2020, doi:10.3390/cancers12040820_

Round 1

Reviewer 1 Report

The authors conducted an NGS-based genetic analyses on 48 EMPD patients, including WGS on 2, RNAseq on 6, WES on 21, and target gene panel sequencing on all patients. The results are very informative. However, the manuscript was not clearly written, difficult to read through. I highly suggest the authors to reorganize the manuscript, make it clearer, and improve the writing.

Organization: 2.1 whole-genome sequencing of two patients with EMPD; 2.2 RNA sequencing of six patients with EMPD; 2.3 whole-exome sequencing of 21 patients with EMPD; 2.4 target sequencing of 48 patients with EMPD or MPD; 2.5 global gene expression profiling of EMPD

Description: case-by-case. In Case 1, describe a fusion was found along with 43 non-synonomous mutations. Then, describe the fusion was verified with RNAseq and RT-PCR. Notably, the authors demonstrated validation result from RT-PCR, but not from RNAseq. After Case 1, describe Case 2 in detail. Describe the results precisely with data, leaving discussion to the Discussion part.

Addition: RNAseq is a great tool to detect fusions, and the authors should have used RNAseq fusion detection for all their RNAseq samples to identify any additional fusions. Mutation detection in RNAseq may also be compared to that in WES and/or target sequencing to see if there are any differences.

Precision: Since no germ-line control was used in WGS or WES, it was not accurate to say these mutations were somatic. Therefore, including more detail of these mutations, compared to hg19 reference, are important for functional analysis.

Details: Criterial (cutoff or threshold) used for mutation detection and gene expression analysis should be described in detail. For supplementary tables, the authors may use Excel to show more detail of these mutations to include these mutations’ potential function or effect, such as clinvar and COSMOS data. The authors should include more detail of DESeq2 results to include expression level and p-value, both of which are critical factors.

Data availability: It will be appreciated that the authors make these NGS data available to public somewhere accessible.

Minor changes: grammar, vocabulary, number, etc.

Reviewer 2 Report

This is an interesting report on molecular and expressional characteristics of extramammary Paget's disease. Unique data are reported from a unique collection of patients with this rare condition. The results are novel and significant.

  • Activating mutations in the FOXA1 gene/promoter are described, but the authors fail to show functional consequences mediated by These mutations. Some functional data of deregulated FOXA1 (or GAS6) would be desirable.
  • This is also desirable for the PIK3CA mutations.
  • Overall, the authors need to show that these mutations have functional consequences.
